# ATTENTION CLUSTERS: REVEALING THE INDUCTIVE BIAS OF ATTENTION MECHANISMS

## ABSTRACT

We introduce a parameter-free framework to isolate the self-attention mechanism, stripping away all learned parameters. Through iterative application, we demonstrate that self-attention alone intrinsically drives the formation of semantically meaningful clusters in the representation space. Analyzing this behavior across global, local-window, and hybrid patterns reveals their inherent geometric biases independent of training. Crucially, we find that query scaling (as used in Longformer) induces an implicit dimensionality reduction that systematically improves model generalization, a insight we validate experimentally. This geometric bias is consistent across both low-dimensional data and high-dimensional real-world representations. Probing a pre-trained model confirms this is architecturally inherent and further refined by learning. Our work provides a useful diagnostic tool for evaluating the relative quality of attention architectures prior to training.

## 1 INTRODUCTION

The Transformer architecture (Vaswani et al., 2017) and its core mechanism—self-attention, underpin modern AI. However, a gap persists between their empirical success and a complete theoretical understanding of their dynamics. Our work connects to and builds upon several research threads. Prior analyses have framed Transformers as dynamical systems (Lu et al., 2019; Geshkovski et al., 2023) characterizing the clustering effect of self-attention in connection to *trained $Q, K, V$* parameters. A key distinction of our work is that we study the attention mechanism in complete isolation from learned parameters, providing an *offline* view of its intrinsic geometric behavior.

We demonstrate that the self-attention mechanism itself iteratively imposes meaningful geometrical structure. The emergent token clusters serve as a visual cue for architectural design and evaluation. With the invention of increasingly cheaper and efficient linear-complexity variants like Longformer(Beltagy et al., 2020), BigBird(Zaheer et al., 2020), etc, to mitigate high computation costs, our framework offers a useful diagnostic to evaluate their representational fidelity relative to the very-costly global attention *before* training, moving beyond trial-and-error design.

The aim here is not to study the bounding geometry of to which the tokens converge, but rather to enable a systematic evaluation of attention mechanisms by tracking the evolution of token representations. Although we utilized low-dimensional data for intuitive visualization, we demonstrate that the insights obtained hereof are not limited to simplified low-dimensional inputs but are a fundamental property of the mechanism, observable even in the high-dimensional hidden states of pre-trained models (see Appendix). We therefore summarize our contributions as follows:

i) We introduce a *parameter-free* setup to iteratively visualize and study the intrinsic clustering behavior of dot-product attention mechanisms.

ii) We analyze and compare the clusters formed by global, local-window, and hybrid (Longformer) attention, revealing their distinct geometric biases.

iii) We uncover a non-obvious architectural detail: query scaling induces an implicit dimensionality reduction, which we hypothesize simplifies the optimization landscape.

iv) We validate this hypothesis experimentally, showing that models with query scaling achieve better downstream generalization. We also probed a pre-trained model to confirm this bias is inherent to the architecture.

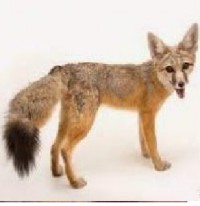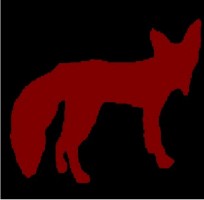

Figure 1: Simple image and corresponding ground-truth mask. The image was resized to $64 \times 64$ pixels and flattened to form the input sequence designated as Data A.

## 2 METHOD

### 2.1 PARAMETER-FREE ATTENTION SIMULATION

We introduce a framework to analyze the geometric transformations induced by dot-product attention. Given an input token sequence $\boldsymbol{X}^{(t)} \in \mathbb{R}^{n \times d}$ at iteration $t$, we apply the update rule:

$$\boldsymbol{Z}^{(t)} = \mathcal{A}(\boldsymbol{X}^{(t)}) \tag{1}$$

$$\widehat{\boldsymbol{Z}}^{(t)} = \frac{\boldsymbol{Z}^{(t)} - \mu}{\sigma} \tag{2}$$

$$\boldsymbol{X}^{(t+1)} := \boldsymbol{X}^{(t)} + \widehat{\boldsymbol{Z}}^{(t)}, \quad \text{for } t = \{0, ..., T\} \tag{3}$$

where, $\mathcal{A}$ is a parameter-free attention mechanism. For our case studies, we utilize standard attention variants from HuggingFace, modified by removing all learnable projections ($\boldsymbol{Q}$, $\boldsymbol{K}$, $\boldsymbol{V}$), layer normalization, and dropout. This isolates the intrinsic behavior of the attention pattern. We evaluate the representational quality of the emergent attention clusters through two complementary metrics.

**Geometric coherence.** Measured by the silhouette score (Rousseeuw, 1987)

$$s(\boldsymbol{x}_i) = \frac{b(\boldsymbol{x}_i) - a(\boldsymbol{x}_i)}{\max\{a(\boldsymbol{x}_i), b(\boldsymbol{x}_i)\}}, \tag{4}$$

for a token $\boldsymbol{x}_i$ assigned to cluster $C_j$, where the cohesion $a(\boldsymbol{x}_i)$ and separation $b(\boldsymbol{x}_i)$ are defined as:

$$a(\boldsymbol{x}_i) = \frac{1}{|C_j| - 1} \sum_{\substack{\boldsymbol{x}_k \in C_j \\ \boldsymbol{x}_k \neq \boldsymbol{x}_i}} \|\boldsymbol{x}_i - \boldsymbol{x}_k\|_2, \tag{5}$$

$$b(\boldsymbol{x}_i) = \min_{l \neq j} \left( \frac{1}{|C_l|} \sum_{\boldsymbol{x}_m \in C_l} \|\boldsymbol{x}_i - \boldsymbol{x}_m\|_2 \right), \tag{6}$$

a high score indicates an architecture with a strong innate bias for forming distinct, compact groups.

**Semantic alignment.** For datasets with ground-truth labels, we measure the semantic meaningfulness of the clusters by calculating the accuracy of a simple cluster-to-label mapping. This involves assigning the most likely label to each cluster based on domain knowledge, and measuring the resulting accuracy against the true labels. A high accuracy indicates that the architecture's clusters aligns with human-defined semantic categories.

### 2.2 DATA PREPARATION

For our analysis, we employ three different input data.

**Simple image.** We construct a simplified input from a less complex RGB image with a homogeneous background resized to $64 \times 64$ pixels (Figure 1) and flattened into a sequence of patch tokens $(1, 4096, 3)$. The initial token distribution of the standardized data is shown in Figure 2a, with the three color dimensions plotted on the x, y, and z axes. We designate this as Data A.

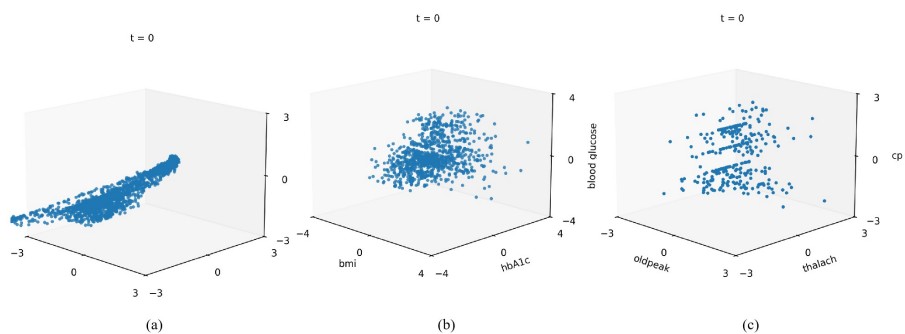

Figure 2: Initial token distribution for the datasets. (a) Simple Image designated as Data A. (b) Diabetes dataset as Data B. (c) Heart disease dataset as Data C.

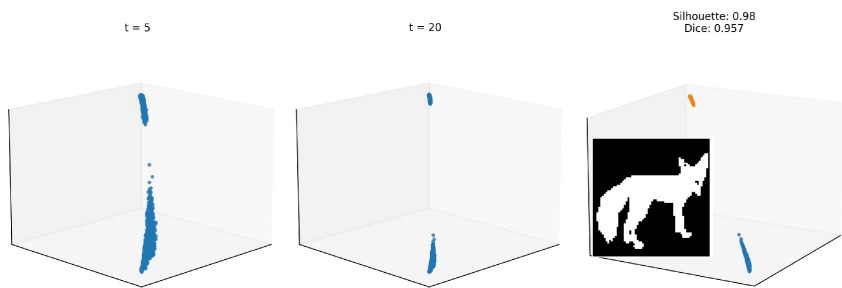

Figure 3: Global attention on Data A forms two coherent clusters (0.98 score) corresponding to the image subject and background, yielding a segmentation mask of dice score of 0.957 when mapped to the ground truth.

**Diabetes dataset.** This dataset (Choksi, 2024) comprises health and demographic data for diabetes-related research. We randomly select 512 samples from each class to obtain a balanced dataset of 1024 samples. We select three features: *bmi* (body mass index), *hbA1c* (hemoglobin level), and *blood glucose*, which are all positively correlated to a diabetic state where {0: non-diabetic, 1: diabetic}. This is designated as Data B, visualized in Figure 2b. (See correlation matrix in Appendix C.1)

**Heart disease dataset.** Obtained from Kaggle (Lapp, 2019), the binary classification task predicts the presence of heart disease where {0: No heart disease, 1: Heart disease}. We slice out a balanced subset of 1024 samples, comprising three features: *oldpeak* (ST depression induced by exercise relative to rest), *thalach* (maximum heart rate) and *cp* (chest pain), where *thalach* is negatively correlated to heart disease and the others have a positive correlation. This is designated as Data C and visualized in Figure 2c.

## 3 THE INDUCTIVE BIAS OF ATTENTION PATTERNS

### 3.1 GLOBAL ATTENTION

Global attention forms tightly grouped, well-separated clusters with minimal fragmentation, a consequence of unconstrained token interactions. On the simple image (Data A), it forms two distinct clusters (Silhouette score 0.98) that correspond precisely to the subject (cat) and the homogeneous background of the image, effectively segmenting the image into its two distinct semantic regions (Figure 3). To visualize this, we assign pure white (255, 255, 255) to all tokens in one cluster and pure black (0, 0, 0) to the other. When reshaped to its original spatial dimensions ($64 \times 64 \times 3$), this assignment produces a binary mask with dice score of 0.957.

Figure 4 shows the clusters for Data B and the corresponding metrics. Here, we encode inter-token attention strength via the thickness of the black lines, revealing that the strongest connec-

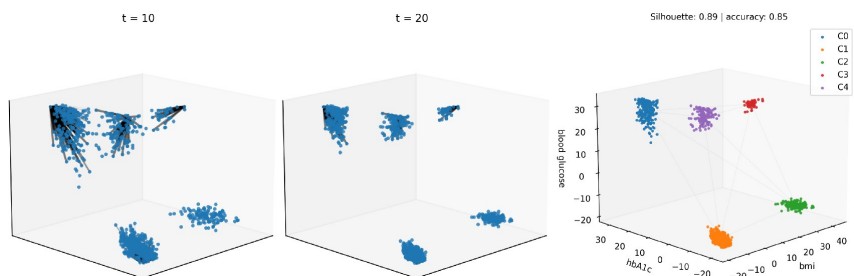

Figure 4: Global attention clusters for Data B shows negligible fragmentation (Silhouette = 0.89, width of the black lines indicate attention strength). Cluster labels {C4: 0, C1: 1} Accuracy score obtains 85% accuracy confirming the content-aware clustering effect of attention.

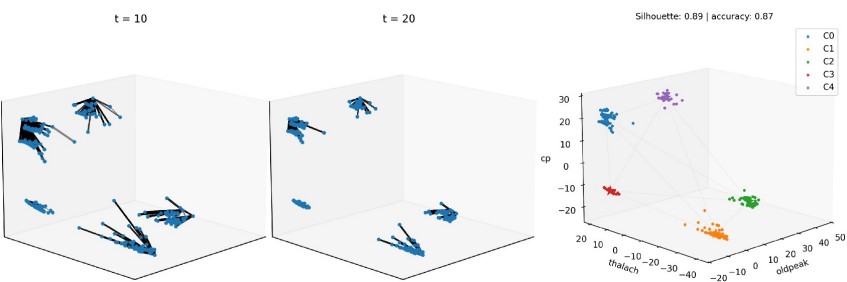

Figure 5: Cluster evaluation of global attention on Data C. A Silhouette score of 0.89 and accuracy of 87% for {C0: 1, C2: 0} quantifies the cluster coherence and semantic meaning.

tions are consistently between *leader tokens* and their cluster members (See Vaswani et al. (2017); Geshkovski et al. (2023) for more details on *leader tokens*). Guided by domain knowledge (*blood glucose*, *hbA1c* and *bmi* correlate to a diabetic state), we assign two cluster labels {C4: 1, C1: 0}. This mapping achieves an accuracy of 85%, strongly validating the semantic structure discovered by attention in isolation from trained parameters. Similarly on Data C (Heart disease dataset), Figure 5 shows the resulting clusters and scores. Based on feature correlations (positive for *cp* and *thalach*, and negative for *oldpeak*), cluster labels {C0: 1, C2: 0} achieve accuracy of 87%.

These results demonstrate that the core mechanics of self-attention can intrinsically impose meaningful geometric structure on data, establishing a foundational organization that learned parameters subsequently refine. This insight enables a systematic framework for evaluating how sparse attention variants alter these inherent clustering properties, revealing the impact of architectural choices on representation geometry, and provides a simple yet useful proxy for inferring the relative quality of an attention mechanism, before committing to the computational cost of full training.

## 3.2 LOCAL-WINDOW ATTENTION

The geometric representations formed by local windows is influenced by two key architectural hyper-parameters: the window size $w$ and the overlap $o$. To isolate the effect of $w$, we analyze cluster formation on Data B (Figure 6a) and Data C (Figure 6b) using non-overlapping windows of sizes $w \in \{128, 512\}$. Visually, we observe significant cluster fragmentation due to the lack of cross-window communication, an effect that is exacerbated with smaller window sizes, directly implying that models employing larger windows would perform better especially in tasks requiring long-range dependencies.

Conversely, for tasks that inherently bias local interactions (e.g., image processing), smaller windows can yield stronger, more precise local features at the cost of global coherence. This is evidenced by the segmentation masks derived from Data A clusters—visual inspection of Figure 7 reveals that $w = 128$ preserves finer local details (particularly evident in the head region) compared to $w = 512$. This also aligns with literature, where local-window Vision Transformers such as Swin Transformer (Liu et al., 2021) achieve state-of-the-art performance even with small ($7 \times 7$) windows.

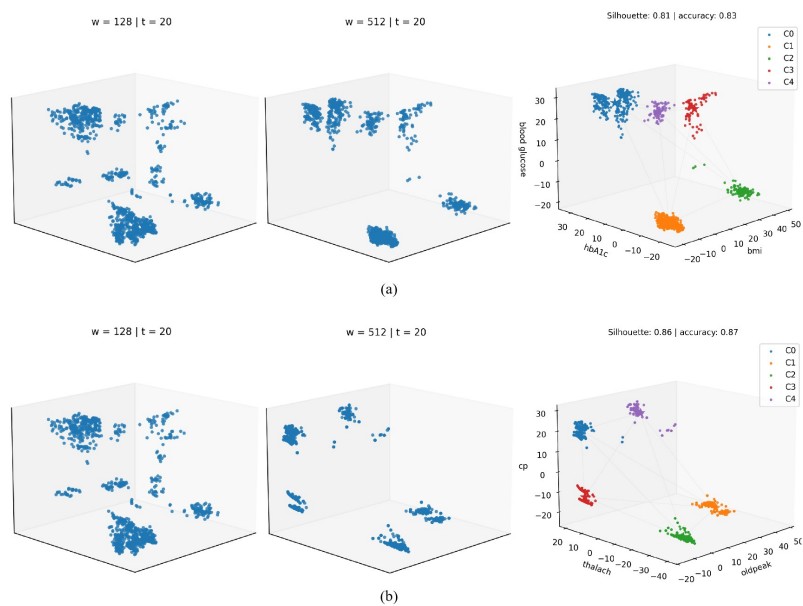

Figure 6: Window size ablated on Data B. Window size $w = 512$ yields more coherent and less fragmented clusters compared to $w = 128$, alluding to the role of $w$ in capturing long-range interactions.

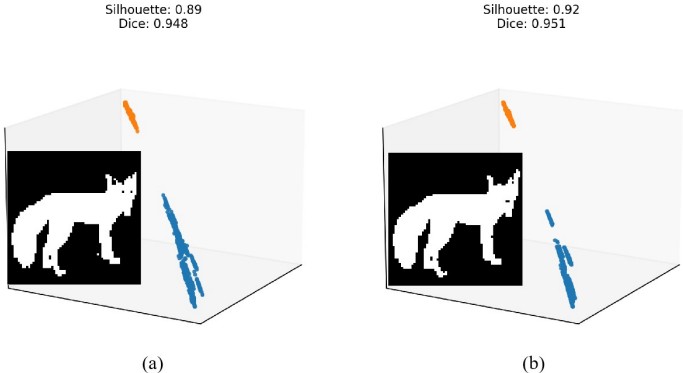

Figure 7: Binary masks derived from clusters on Data A with (a) $w = 128$ and (b) $w = 512$. The mask for $w = 128$ preserves finer local features (e.g., details in the head).

Compared to global attention, the cluster metrics for local-window attention are significantly lower, showing that despite its linear complexity advantage, the isolated nature of non-overlapping windows lead to fragmented representations, underscoring the need for mechanisms that facilitate interaction between windows. Hybrid approaches, such as the Longformer address this limitation by augmenting local windows with global tokens, dilated windows and overlapping regions to efficiently incorporate global dependencies.

## 3.3 LONGFORMER

The Longformer employs a hybrid attention pattern combining overlapping local windows, dilated windows, and a subset of globally attended tokens:

$$A = \underbrace{A_{\text{local}}}_{\text{Window}} + \underbrace{A_{\text{global}}}_{\text{Task tokens}} + \underbrace{A_{\text{dilated}}}_{\text{Long-range}}$$

Consequently, its cluster representation depends on window properties, dilation rate, and the selection of global tokens. However, the standard HuggingFace implementation omits dilation due to its

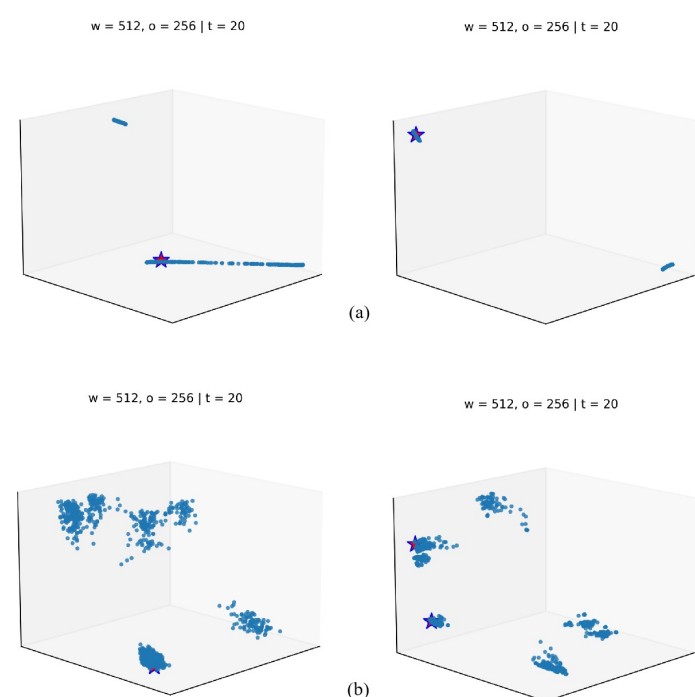

Figure 8: Attention clusters on Data B (first position as global token) and Data C (first and last positions as global tokens) for Longformer variants. (a) With query scaling, tokens cluster towards a plane. (b) With standard dot-product scaling, tokens cluster towards vertices. Red stars are global tokens.

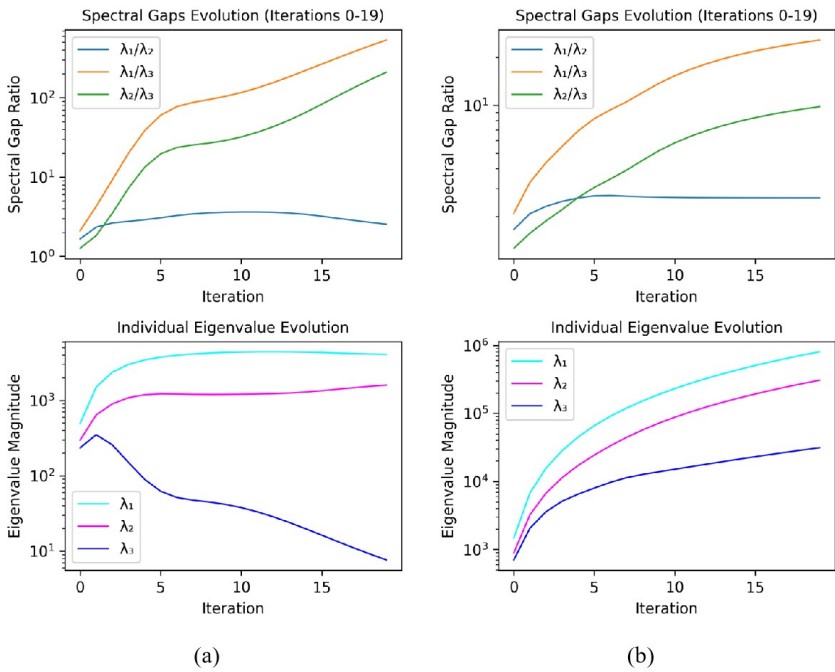

Figure 9: Spectral analysis. (a) Query scaling induces a widening spectral gap, suppressing some eigenvalues and effectively reducing the feature space dimensionality. (b) Standard dot-product scaling did not produce this widening effect.

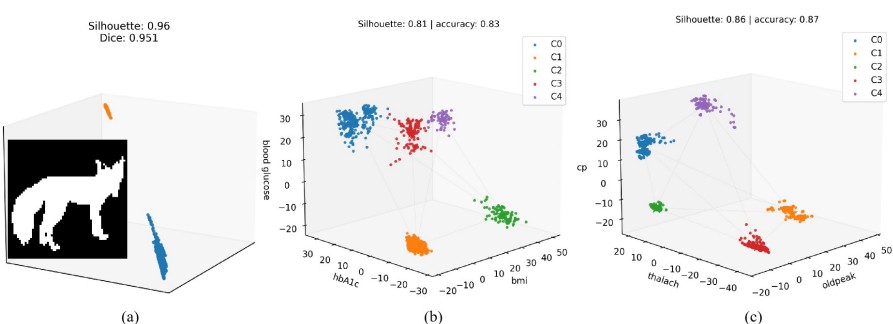

Figure 10: Cluster evaluation for Longformer SDP variant for (a) Data A, (b) Data B, and (c) Data C. Visual inspection confirms a reduction in cluster fragmentation attributed to cross-window interactions from overlapping windows and global tokens

requirement for custom CUDA kernels. Therefore, our analysis focuses on the combined effect of overlapping windows and global tokens, and other architectural choices on the resulting attention clusters.

A unique detail we observed in the HuggingFace's Longformer implementation is its use of query scaling rather than scaling the full dot-product by $1/\sqrt{d_k}$ as prescribed by (Vaswani et al., 2017). We find this design choice has a profound and mechanistic impact on the geometry of the resultant attention clusters. Figure 8 visualizes the respective clusters on Data B (left) and C (right) for case (a) scaled query and (b) scaled dot-product. In case (a), the tokens appear to cluster towards planes, similar to the *clustering towards hyperplanes* phenomenon observed in (Geshkovski et al., 2023) when the matrices $(\boldsymbol{Q}, \boldsymbol{K}, \boldsymbol{V})$ form a *good triple*. As our setup is agnostic to these weight matrices, we directly analyzed the eigenvalues of the input's covariance matrix at each iteration. Our empirical analysis reveals that scaling the queries induces a large spectral gap between the leading eigenvalues and the rest (Figure 9), in effect implicitly reducing the dimensionality of the embedding space. We hypothesize that this implicit dimensionality reduction could accelerate convergence and generalization on downstream tasks by simplifying the optimization landscape—we verify this by experiment in Section 4.

However, for direct comparison with the clusters of global and local windows attention, the results in Figure 10 focus only on the SDP variant. The attention clusters for Data A using window size $w = 512$ (global token selected as middle position) obtains a Silhouette of 0.96, 4-points higher than a local window attention and 2-points less than global attention. Although the evaluation metrics for (b) Data B and (c) Data C, are similar to those of local windows, visual inspection reveals noticeable reduction in cluster fragmentation, highlighting the effect of cross window interactions via overlap and global tokens. This implies that Longformer's hybrid is a relatively closer approximation to global attention than isolated local windows for a given $w$.

## 4 EXPERIMENTAL VALIDATION

While the superiority of hybrid attention over simple local windows is well-established (Beltagy et al., 2020; Zaheer et al., 2020), our goal is to validate the insights from Section 3.3 by comparing the generalization performance of the scaled query (SQ) and scaled dot-product (SDP) variants of the Longformer architecture.

### 4.1 SETUP

To remain within computational constraints, we use a small-scale dataset WikiText-2 (Merity et al., 2016) and a scaled-down model architecture . We pre-train each variant from scratch using masked language modeling (MLM) for only 80 epochs and finetune it on IMDB Reviews (Maas et al., 2011) and SST-2 (Wang et al., 2019) datasets for 10 epochs (See Appendix C for full details). Note that our aim is not to achieve state-of-the-art performance but to validate the insights derived from our attention cluster analysis.

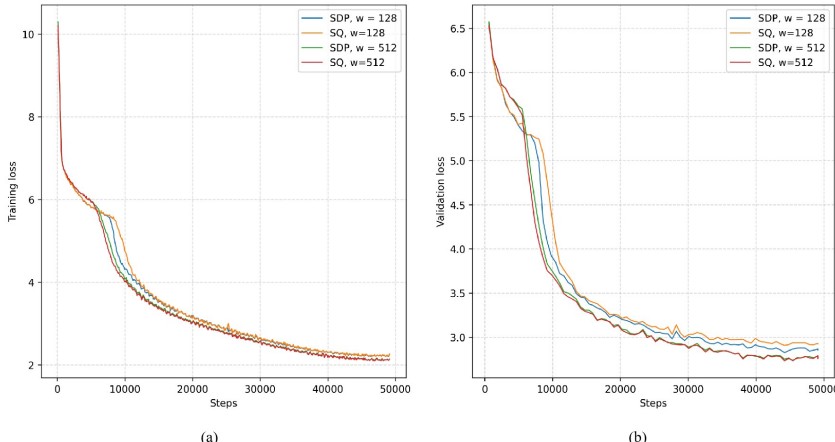

(a)                                             (b)

Figure 11: MLM pre-training losses for different scaling strategies and window sizes. The SDP variant achieves marginally lower losses on the pretext task, while the SQ variant demonstrates better downstream generalization (see Table 1)

Table 1: Fine-tuning results on IMDB and SST-2 benchmark. The SQ variant consistently outperforms the SDP variant across both downstream tasks, demonstrating better generalization (accuracy/F1 scores averaged across two runs).

| Model Variant | Pre-training | IMDB | SST-2 |
|---|---|---|---|
| SDP, $w = 128$ | 2.253 | 88.0 / 87.9 | 80.6 / 80.6 |
| SQ, $w = 128$ | 2.267 | **88.8 / 88.9** | **83.1 / 83.8** |
| $\Delta$ (SQ - SDP) | | +0.8 / +1.0 | +2.5 / +3.2 |

## 4.2 RESULTS

Figure 11 compares the training and validation losses during pre-training. While the SDP variant achieves marginally lower pre-training loss, suggesting slightly better optimization on the pretext task, the SQ variant demonstrates superior generalization on the downstream sentiment analysis task, obtaining higher accuracy and F1 scores (Table 1). This result aligns with our hypothesis from Section 3.3—the implicit dimensionality reduction induced by query scaling simplifies the feature space, which can act as a regularizer and improve generalization on the downstream tasks.

## 4.3 VALIDATION IN TRAINED MODELS

To show that the SQ effect is also fundamental to trained models and not just an artifact of our simulation, we probe a pre-trained Longformer-base. We analyze its hidden-state eigenvalues across all layers using a real text input from the IMDB validation set. First, with layer normalization enabled and all learned weights disabled, the SQ attention mechanism alone induces a strong spectral gap (Figure 12), similar to our parameter-free result from Figure 9a, proving the bias is architectural. Second, activating the trained $(\mathbf{Q}, \mathbf{K}, \mathbf{V})$ matrices refines this pre-existing bias (Figure 13), sharpening the low-rank structure.

This provides a mechanistic explanation for our empirical results that the variant's superior generalization stems from its architecturally pre-simplified learning landscape, which the optimizer further refines and amplifies. In addition, it confirms our offline analysis as a powerful diagnostic and comparative tool for predicting the relative behavior of a model prior to training.

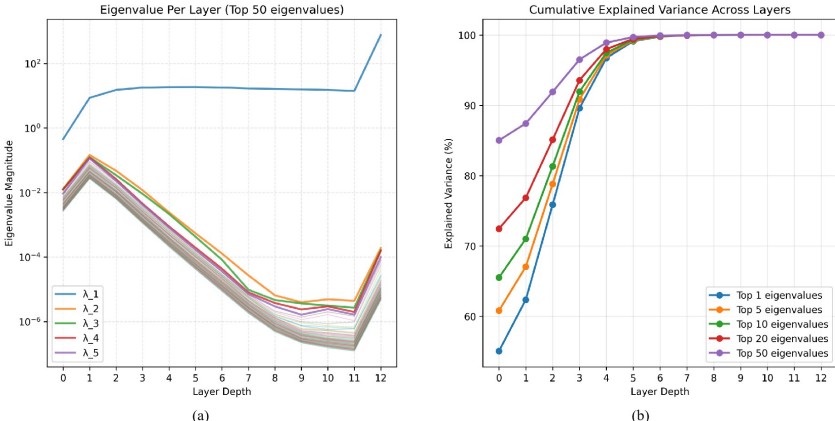

Figure 12: Eigenvalues with SQ and layer normalization active, while learned $(\boldsymbol{Q}, \boldsymbol{K}, \boldsymbol{V})$ weights are disabled. The wide spectral gap is similar to our parameter-free result, proving the effect of the SQ architecture.

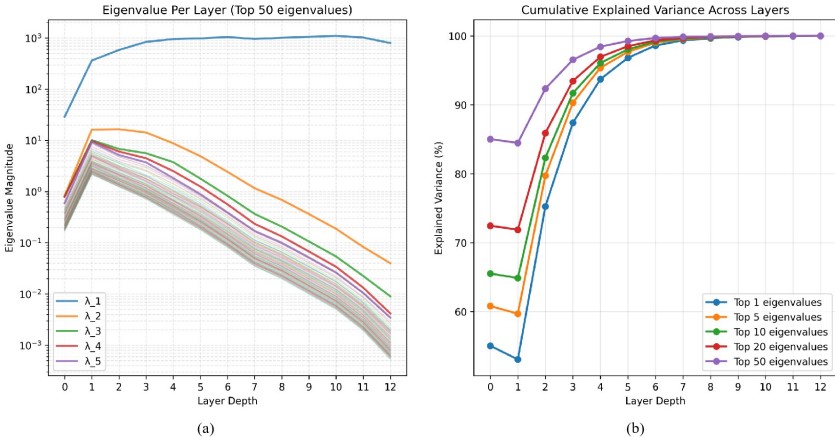

Figure 13: Eigenvalues with both SQ, layer normalization and learned $(\boldsymbol{Q}, \boldsymbol{K}, \boldsymbol{V})$ weights active. The trained matrices refine the pre-existing architectural bias, sharpening the low-rank structure.

## 5 CONCLUSION

Our work establishes attention clusters as a powerful diagnostic tool for attention mechanism analysis, providing a lens for rapid architectural assessment, validating efficient approximations, and uncovering non-obvious mechanistic details, such as the generalization benefits of query scaling.

However, this study has a limitation that points toward a productive future research direction. This work provides a static, *offline* view of attention's inductive bias. A compelling extension would be to investigate the relationship between these parameter-free clusters and the dynamics of actual model training to see if they act as attractors for the learning process. By bridging geometric analysis with architectural design, this line of research moves beyond trial-and-error ablations and offers a path toward transformers informed by their intrinsic representational properties.

## 6 LLM USAGE

The AI research assistant Claude (Anthropic) was used in refining the text and providing constructive feedback during the preparation of this manuscript.

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

# A   MORE ANALYSIS OF QUERY SCALING IN PRE-TRAINED MODELS

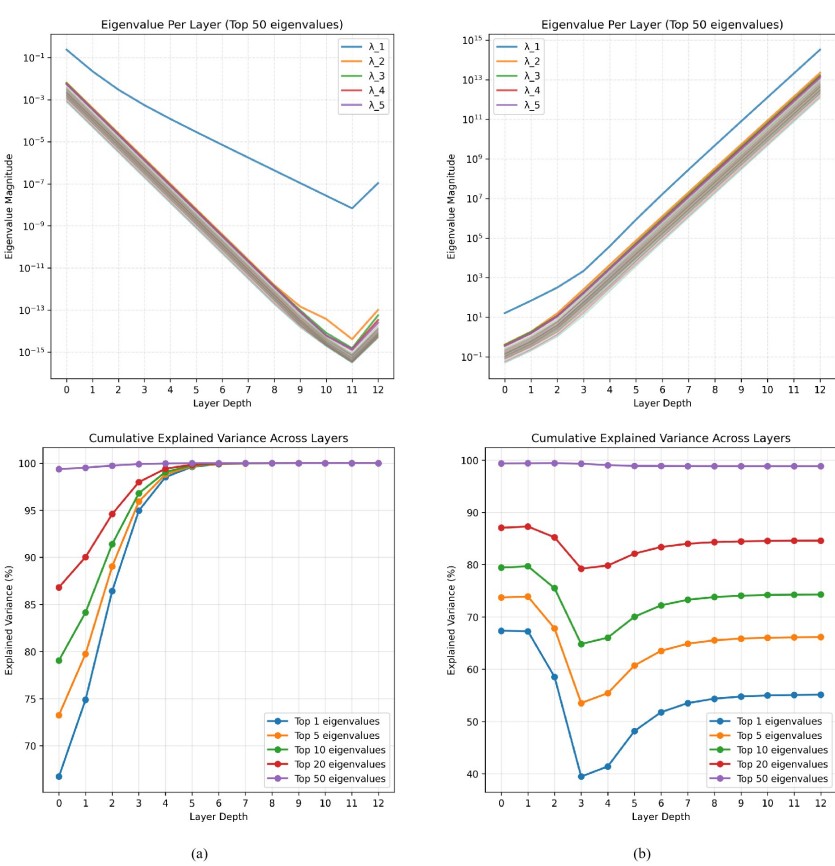

(a)                                                    (b)

Figure 14: Spectral analysis with all learned parameters disabled for (a) Longformer (SQ) and (b) Roberta (SDP). The SQ architecture intrinsically drives a low-rank representation.

We provide further validation of our spectral analysis by comparing a pre-trained *Longformer-base* (SQ) model against a pre-trained *Roberta-base* (SDP) model. This further verifies that the low-rank bias induced by query scaling is not an artifact of our parameter-free simulation but is present in and amplified by trained models. Both models were probed with the same input text.

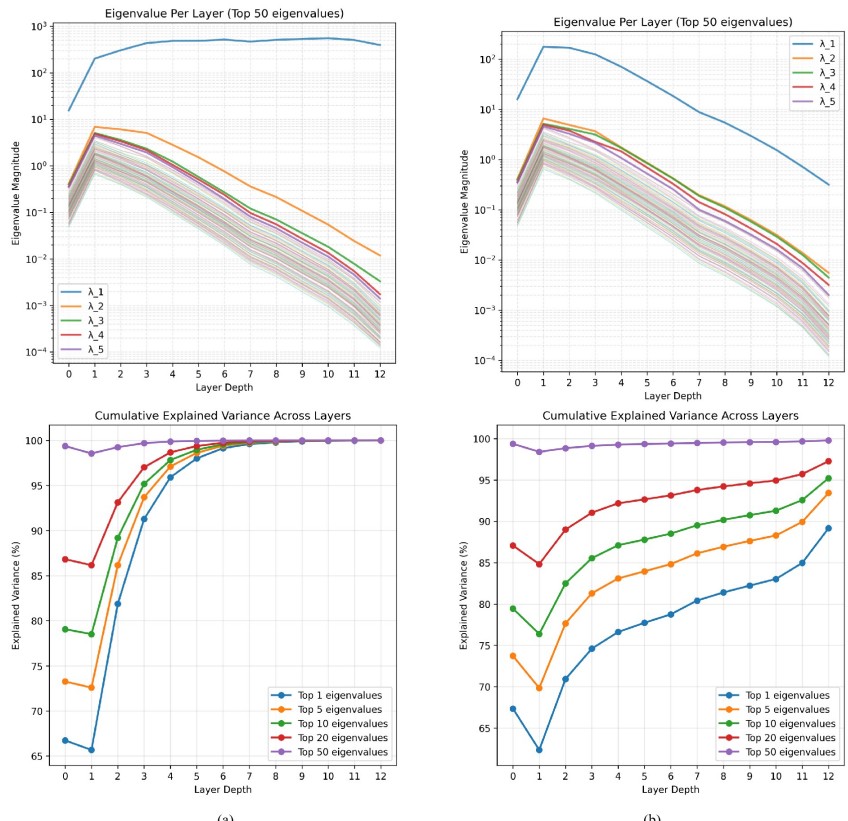

Figure 15: Spectral analysis with full models active for (a) Longformer (SQ) and (b) Roberta (SDP). Learning amplifies the inherent low-rank bias of the SQ architecture.

---

**Input text**

> As much as the movie was good, i have nothing more to say about it
> than what was said already.  all i wanted is to point the fact that
> the movie isnt from Sweden but from Denemark.  Maybe I wrong and in
> that case i'll be happy to know my mistakes so take the and notify
> me.

---

**With parameters disabled.** Figure 14 compares the eigenvalue plots with all learned parameters and normalization disabled. The results from our offline simulation (Section 3.3) are directly replicated: the SQ architecture alone induces a wide spectral gap, with the leading eigenvalue dominating the variance by the middle layers. In contrast, the SDP variant preserves more activity across the spectrum, demonstrating a weaker inherent low-rank bias.

**With parameters enabled.** Figure 15 shows the eigenvalue spectra with the attention parameters active ($\mathbf{Q}, \mathbf{K}, \mathbf{V}$, and layer normalization). The learning process refines and *amplifies* the pre-existing architectural bias. The spectral gap in the SQ model becomes even more pronounced, confirming that training sharpens the inherent low-rank structure. This provides a mechanistic explanation for the SQ variant's superior generalization performance—it optimizes on an architecturally pre-simplified landscape.

This analysis solidifies our core claim in Section 3.3: query scaling is an architectural choice that imposes a low-rank inductive bias, which is subsequently exploited and enhanced during training to achieve better generalization. It also suggests that the framework's insights are scalable and applicable to the complex, high-dimensional spaces where transformers are actually deployed. The

"simplified" 3D analysis serves as an intuitive visual proof of concept, while the high-dimensional spectral analysis validates its real-world relevance.

## B   ATTENTION CLUSTERS AS A TOOL FOR ARCHITECTURE DESIGN

Our parameter-free attention cluster framework offers a powerful methodology for accelerating the development of new efficient transformer architectures. Traditionally, designing novel attention patterns involves a costly cycle of implementation, full-scale training, and evaluation—a process that is computationally prohibitive and time-consuming.

Our approach provides a rapid and inexpensive proxy for this process. Researchers can now:

- a) Propose a new sparse or efficient attention pattern.
- b) *Instantaneously* evaluate its intrinsic representational properties using our offline setup.
- c) Compare the resulting clusters and metrics (e.g., Silhouette score, fragmentation) directly against a global attention baseline and other existing efficient variants (e.g., Longformer, BigBird).
- d) Only commit the resources to *train* the most promising candidates—those whose offline cluster structure most closely approximates the desired global attention geometry.

**Case study: mitigating small-window fragmentation.**   For instance, a model using small, non-overlapping windows is cheap but suffers from high fragmentation and unsuitable for tasks that require coherent global relationships (Section 3.2). To improve it, a designer might propose mechanisms like shifted windows, dilated attention, or adding global tokens (or any other novel ideas). Instead of training each variant to convergence, our framework allows for a quick offline evaluation of these ideas:

> **A designer may ask?**
>
> For my proposed new pattern $\mathcal{A}$, how does its cluster coherence and fragmentation at window size w = 64 compare to that of a standard local window and a global attention baseline on my target dataset?

The qualitative visualizations and quantitative metrics provide an immediate signal. A variant that reduces fragmentation and yields a higher Silhouette score is a strong candidate for successful training. This moves architectural development beyond costly trial-and-error, towards a more principled and efficient design loop centered on understanding an architecture's inherent representational bias before any learning occurs.

## C   EXPERIMENT DETAILS

Table 2: Model configuration

| Model configuration | Value |
| --- | --- |
| Hidden dimension | 256 |
| Attention heads | 4 |
| Intermediate size (FFN) | 1024 |
| Number of layers | 6 |
| Max sequence length | 1024 |
| Window size | 128 |

The model configuration, pre-training/finetuning parameters for the experiments in Section 4 are given in Table 2 and Table 3 respectively. The pre-training dataset WikiText-2, was subsampled by filtering sequences shorter than 512 tokens, resulting in approximately 9.8K training and 1K validation samples. We used a dynamic masking following Liu et al. (2019).

Table 3: Pre-training and finetuning parameters

| Hyper-parameters | Pre-training | Finetuning |
|---|---|---|
| Batch size | 16 | 8 |
| Learning Rate | $9 \times 10^{-4}$ | $2 \times 10^{-5}$ |
| Epochs | 80 | 5 |
| Warm-up steps | 10,000 | — |
| Weight decay | $2 \times 10^{-6}$ | — |
| Window size | 128 | |
| Masking ratio | 15% | — |

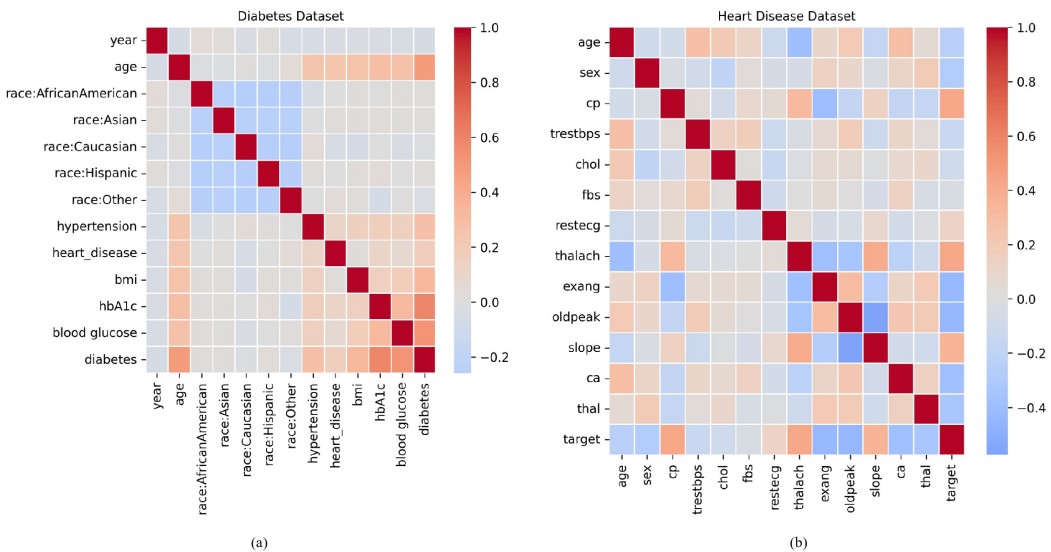

(a)   (b)

Figure 16: Feature correlation analysis for medical datasets. (a) Diabetes dataset: selected features (*blood glucose, HbA1c, BMI*) all show positive correlation with diabetes outcomes. (b) Heart disease dataset: selected features show mixed correlations - positive for *cp* (chest pain) and *thalach* (max heart rate), but negative for *oldpeak*

### C.1   CORRELATION MATRIX FOR DATASETS

The correlation matrix for the Diabetes dataset (Data B) and the Heart disease dataset (Data C) in Section 2.2 is visualized in Figure 16.

## D   EXTENDED DISCUSSION OF RELATED WORK

Our work on the intrinsic geometric bias of attention sits at the intersection of several research areas. We expand on the connections briefly outlined in the introduction.

### D.1   TRANSFORMER EXPLAINABILITY AND DYNAMICS

Recent work has sought to demystify the Transformer's internal workings through powerful mathematical frameworks. A prominent line of research models transformer dynamics as systems of interacting particles (Lu et al., 2019; Sander et al., 2022; Geshkovski et al., 2023). These studies have been instrumental in characterizing phenomena like clustering in *trained* models. Complementary to these, visualization-based approaches (Chefer et al., 2021; Fantozzi & Naldi, 2024; Abnar & Zuidema, 2020) provide valuable per-instance insights into model decisions.

We bridge these perspectives by providing a *parameter-free* simulation framework. By stripping away all learned parameters, we isolate the core attention mechanism to study its *intrinsic*, pre-learning geometric properties. This offers an *offline* diagnostic tool that reveals the inductive bias of the architecture itself, before any training occurs.

## D.2 CONNECTION TO NEURAL COLLAPSE

The geometric structure of the clusters formed by our framework bears a striking resemblance to the *neural collapse* (NC) phenomenon (Papyan et al., 2020). NC describes the tendency of deep classifiers to learn features where within-class variability collapses to a single point and class means form a symmetric simplex.

We demonstrate that a form of collapse is not merely an endpoint of supervised learning but is an *innate driving force* of the self-attention mechanism itself. Given labeled data, self-attention iteratively structures the representation space such that points of the same class converge, even in the complete absence of learning. This suggests that neural collapse may be, in part, architecturally predisposed by attention.

## D.3 IMPLICATIONS FOR EFFICIENT TRANSFORMER VARIANTS

The quadratic complexity of standard self-attention has spurred immense innovation in efficient approximations, including sparse patterns (e.g., Longformer (Beltagy et al., 2020), BigBird (Zaheer et al., 2020)), linear attention (e.g., Performer (Choromanski et al., 2020), Linformer (Wang et al., 2020)), and kernel-based methods (Katharopoulos et al., 2020).

Our framework provides a new lens to quickly evaluate subsequent architectures by their *representational fidelity*. By comparing the clusters formed by an efficient variant to those of global attention, we obtain an abstract measure of its approximation quality *prior to training*. This moves the evaluation of efficient transformers beyond just FLOPs and wall-clock time, towards a more nuanced understanding of how their architectural choices alter the fundamental geometry of the representation space.

