# OpenReview forum: "Attention Clusters: Revealing the Inductive Bias of Attention Mechanisms"
_ICLR.cc/2026/Conference — ICLR 2026 Conference Withdrawn Submission_

### Official Review · Reviewer_tD5A · 2025-10-31

**Soundness:** 2
**Presentation:** 2
**Contribution:** 2
**Rating:** 2
**Confidence:** 4

**Summary:**

The paper proposes a parameter-free framework to isolate self-attention by stripping all learned projections and layers, then iteratively applying attention to inputs to visualize emergent attention clusters. Using a toy 64×64 RGB image and two small tabular datasets (diabetes, heart disease), the authors report high silhouette scores and semantic alignment of clusters. They also compare global, local-window, and Longformer-style hybrid attention, and claim that query scaling (in Longformer) induces an implicit dimensionality reduction evidenced by spectral gaps and slightly better downstream generalization on tiny text benchmarks after brief pretraining.

**Strengths:**

1. Side-by-side visualizations for global vs. local-window vs. Longformer attention and a concrete experiment comparing scaled-query (SQ) vs scaled dot-product (SDP) variants.

**Weaknesses:**

1. Incomplete and unclear writing: Several core elements such as motivation, related work, and key definitions, are missing or underdeveloped, which makes the paper hard to read and follow.
2. Conceptual triviality / unclear novelty: With all projections and normalizations removed, self-attention effectively reduces to a (softmax-normalized) dot-product kernel over raw tokens; the update rule then corresponds to repeated kernel-weighted averaging (plus standardization), which is closely related to classical affinity/spectral clustering behavior. Many reported “properties” (e.g., cluster formation) are thus properties of the dot product kernel, and not new insights about trained attention. The paper does not connect its dynamics to known kernel/spectral results nor clarify what is genuinely new beyond prior analyzes of clustering in self-attention dynamics.
3. Toy-level evaluation: The semantic alignment relies on manual cluster-to-label mapping guided by domain knowledge, which creates room for post-hoc choices. The datasets are tiny and low-dimensional, and the image dataset (dataset A) is a trivial color-based foreground/background split.

**Questions:**

1. First, on weakness #1, the paper seems to be rushed and mostly LLM-generated (although acknowledged) without further manual improvement/validation. For example, the introduction is very short and the motivation of the work is not clearly mentioned. Furthermore, the paper is presented in a very complicated and ambiguous way that is hard to follow. For example, several concepts (such as local window in section 3.2) were mentioned without proper explanation. Could you please explain the motivations of the work, considering the questions below?
2. On weakness #2, let's start with the analyzed framework. It is mentioned that all learnable projections, normalizations, and dropouts are removed. This means that self-attention would exactly calculate the (normalized) dot product (i.e. similarity) of different tokens in the input. In that case, for data A (the image of a fox), this will correspond to a color-based clustering of the image which makes the reported results largely trivial. I would guess the same concern also applies to other datasets. Could you please justify how the analyzed framework goes beyond dot-product properties?
3. I would appreciate it if the authors could explain what the novelties of the paper are. In particular, under what conditions does the proposed parameter-free self-attention differ (both theoretically and empirically) from standard spectral clustering [1] or diffusion maps [2] on the cosine kernel? Concrete derivations or counterexamples would help.
4. Assuming that the abovementioned concerns are addressed, I would like to see the results on high-dimensional hidden states from diverse pretrained models (not just Longformer) and on non-trivial vision inputs beyond simple foreground/background segmentation.


[1] Von Luxburg, Ulrike. "A tutorial on spectral clustering." _Statistics and computing_ 17.4 (2007): 395-416.

[2] Coifman, Ronald R., and Stéphane Lafon. "Diffusion maps." _Applied and computational harmonic analysis_ 21.1 (2006): 5-30.

---

### Official Review · Reviewer_3Lb4 · 2025-10-31

**Soundness:** 1
**Presentation:** 1
**Contribution:** 1
**Rating:** 2
**Confidence:** 4

**Summary:**

The paper proposes an offline, parameter-free framework to “reveal” the inductive bias of different self-attention mechanisms. Concretely, the authors take standard attention modules (BERT self-attention, Longformer self-attention, and a custom local-window attention), remove their learnable parts by replacing the query/key/value projections with identity matrices, and repeatedly apply these frozen attention layers to an input tensor. After each iteration, they cluster the resulting token representations with k-means and evaluate how “clean” the clusters are using silhouette score and simple visualizations (3D scatter, binary masks). The claim is that different attention patterns (global vs local vs Longformer-style) induce different clustering behaviour even without training, and that this can be used as a diagnostic tool for attention designs. Experiments are run on three very small setups: (1) a single 64×64 image (probably a fox, not a cat) with very very clear foreground/background separation, (2) a 3D diabetes tabular dataset, and (3) a 3D heart-disease dataset. There is also a small NLP pretraining/fine-tuning experiment to support the claim that a particular Longformer scaling variant (“sq” vs “sdp”) generalizes better.

**Strengths:**

1. **Code-level clarity.**
2. **Didactic value.** As a teaching/demo tool (“what does attention do if you strip away learning?”) this is nice. The visualizations of token trajectories and the k-means masks can help beginners understand that attention is, at its core, a data-dependent mixing operator.
3. **Interesting small observation on Longformer.** The paper notices that the HF Longformer variant scales only queries (scaling_type='sq') and shows that this has an effect on the spectrum and the resulting clusters. That’s a legitimate engineering observation.
4. **Connection (at least in intent) to recent works on clustering in attention dynamics.** The authors mention the line of work that interprets attention as a particle system / clustering dynamic. So they are not completely disconnected from the literature — they just don’t reach the same depth.

**Weaknesses:**

1. **Experimental setup is far below ICLR standards.** The main vision experiment is literally one image of size 64×64, with a very obvious foreground (fox) and background. On such an image, plain k-means on raw RGB would already separate fg/bg. So this setup does not demonstrate any special, emergent, or surprising property of attention. The other two datasets are tiny 3D tabular datasets (diabetes, heart). In 3D it is trivial to see clusters and to make nice 3D plots. That does not tell us how the method behaves in realistic, high-dimensional vision or language settings. For a paper that claims to “reveal” something about attention, we would minimally expect CIFAR-10, CUB, Stanford Cars, or at least a few real images passed through an attention mechanism.
2. **No coherent rationale for the chosen attention mechanisms**. The paper only uses:
	•	“global” = BERT self-attention (but with identities),
	•	Longformer self-attention (with identity Q/K/V),
	•	a custom local-window attention.
But the motivation in the text is to study different attention patterns. If that’s the goal, please include more (possibly self-attention or cross-attention mechanisms). Have a look at [1] for various attention mechanism alternatives.
3. **Metrics are weak and partly circular**.
4. **The core claim is obvious and not backed by strong evidence**. The conclusion — “global attention gives more coherent clusters, local attention is fragmented, Longformer is in-between” — is exactly what one would predict from the definitions of these mechanisms, without running any experiments. The experiments should uncover a non-obvious behaviour (e.g. “this sparse pattern still recovers semantic parts”, or “this scaling makes clustering drastically more stable”), but they don’t.
5. **Very thin theoretical background**.
6. **The NLP experiment does not support the big story**. Authors add a small WikiText-2 → IMDB/SST-2 experiment to argue that “the scaling we identified offline corresponds to better generalization”. But the gains are small, there’s no variance, and the setup is small. This does not validate the proposed analysis pipeline.
7. **Missing related works**. [2], [3]

[1] Psomas, Bill, et al. "Keep it simpool: Who said supervised transformers suffer from attention deficit?." Proceedings of the IEEE/CVF International Conference on Computer Vision. 2023.
[2] Edelman, Benjamin L., et al. "Inductive biases and variable creation in self-attention mechanisms." International Conference on Machine Learning. PMLR, 2022.
[3] Mijangos, Víctor, et al. "Relational inductive biases on attention mechanisms." arXiv preprint arXiv:2507.04117 (2025).

**Questions:**

Authors could possibly do the following actions to make this paper publishable:
- Provide more theoretical analysis
-  Broaden datasets
-  Broaden attention variants
- Show a real use case
- Clean up the writing and presentation (visualizations, etc.)

---

### Official Review · Reviewer_TFXB · 2025-11-01

**Soundness:** 1
**Presentation:** 1
**Contribution:** 1
**Rating:** 2
**Confidence:** 4

**Summary:**

The paper proposes a “parameter-free” approach to study self-attention's supposed intrinsic geometry. All learnable components are explicitly removed from a few LLMs, leaving only repeated applications over attention of dot-products. The authors then visualize the resulting “attention clusters” for a few attention types, including Longformer on arbitrary inputs, and claim that this reveals an inductive bias of attention.  They also compare scaled query vs. scaled dot-product attention in both parameter-free and trained settings, arguing that SQ introduces an implicit low-rank bias that improves generalization. Some experiments are performed to argue this.

**Strengths:**

1. The motivation is interesting
2. The observations of clustering aligning with semantics of input data is surprising

**Weaknesses:**

1. The approach itself is questionable: repeatedly applying a parameter-free attention mechanism may not be informative about trained model behavior.
2. The fact that clustering of param-free models aligns with semantic labels indicates that the tasks considered were too simple.
3. SQ vs SDP comparisons are unclear, no statistical significance is reported, evals are done only for two datasets
4. The link between param-free and actual trained models' behavior remains unestablished, in my opinion.

**Questions:**

See Weaknesses

---

### Official Review · Reviewer_vqCG · 2025-11-05

**Soundness:** 3
**Presentation:** 3
**Contribution:** 3
**Rating:** 6
**Confidence:** 3

**Summary:**

The paper presents a parameter-free framework for analysing the inductive bias of self-attention mechanisms. By removing all learned parameters, the authors show that attention alone can produce semantically meaningful clusters and reveal distinct geometric biases across different attention types. They also identify that query scaling, as used in the Longformer, introduces an implicit dimensionality reduction that may improve generalisation.

This is an original and well-written paper that offers a creative approach to understanding the intrinsic properties of attention mechanisms. The idea of using a parameter-free setup is appealing, and the insights about query scaling are thought-provoking. However, the evidence is somewhat limited in scope, and the presentation could be improved, particularly in the visual materials.

**Strengths:**

-	Conceptually novel and well motivated. The framework isolates the intrinsic geometry of attention, offering a new way to study the mechanism itself.
-	Clear technical presentation with well-defined update rules and evaluation metrics.
-	The finding that query scaling induces a low-rank bias is original and backed by experiments.
-	Bridges theoretical and practical perspectives, providing a potential diagnostic tool for model design.
-	The paper is clearly structured and generally easy to follow.

**Weaknesses:**

-	The visualisations are bit unclear. The 3D scatter plots and segmentation examples are conceptually useful but could be presented a bit better. Figure 3 refers to a “cat” even though the image depicts a fox, which creates a small but distracting inconsistency.
-	The evaluation does not include modern or large-scale models. The experiments rely on older architectures such as the Longformer and RoBERTa, and on toy datasets. The framework is claimed to generalise to high-dimensional data, but this is not demonstrated with models such as DINOv2, CLIP, or even modern efficient Transformers (e.g., Mamba, FlashAttention2).
-	The experimental validation is small and lacks statistical analysis. The reported improvements in downstream accuracy and F1 score are minor and not tested for significance.
-	Limited connection to explainability research. Although the framework is described as a diagnostic tool, it does not fully engage with the wider literature on interpretability or explainable AI.

**Questions:**

N/A

---

### Note · Authors · 2025-11-28

I have read and agree with the venue's withdrawal policy on behalf of myself and my co-authors.